# p53 and p21 dynamics encode single-cell DNA damage levels, fine-tuning proliferation and shaping population heterogeneity

Nica Gutu[1,2], Neha Binish [1,5], Ulrich Keilholz[1,3], Hanspeter Herzel [1,2,4] & Adrián E. Granada [1,3✉]

Cells must accurately and quickly detect DNA damage through a set of checkpoint mechanisms that enable repair and control proliferation. Heterogeneous levels of cellular stress and noisy signaling processes can lead to phenotypic variability but little is known about their role in underlying proliferation heterogeneity. Here we study two previously published single cell datasets and find that cells encode heterogeneous levels of endogenous and exogenous DNA damage to shape proliferation heterogeneity at the population level. Using a comprehensive time series analysis of short- and long-term signaling dynamics of p53 and p21, we show that DNA damage levels are quantitatively translated into p53 and p21 signal parameters in a gradual manner. Analyzing instantaneous proliferation and signaling differences among equally-radiated cells, we identify time-localized changes in the period of p53 pulses that drive cells out of a low proliferative state. Our findings suggest a novel role of the p53-p21 network in quantitatively encoding DNA damage strength and fine-tuning proliferation trajectories.

[1] Charité Universitätsmedizin, Charité Comprehensive Cancer Center, Berlin, Germany. [2] Humboldt-Universität zu Berlin, Berlin, Germany. [3] German Cancer Consortium, Deutschen Konsortiums für Translationale Krebsforschung (DKTK), Berlin, Germany. [4] Institute for Theoretical Biology, Berlin, Germany. [5]Present address: Hertie Institute for Clinical Brain Research, Center for Neurology, University Medical Center Tübingen, Tübingen, Germany. ✉email: adrian.granada@charite.de

The development of effective cancer treatments is hampered by the variability of tumor cell subpopulations and their divergent patterns of resistance and growth. A better understanding of the processes driving tumor heterogeneity is key to the development of new treatment strategies. Phenotypic within-tumor growth differences are routinely quantified through standardized methods such as immunohistochemistry staining of proliferation markers (e.g., Ki67, PhosphoRB), cell-replating assays, or estimations of cell-population growth through ATP-based viability assays[1,2]. Bulk metrics, such as tumor proliferation rate, are widely used in clinical practice to determine tumor aggressiveness and guide treatment decisions. Highly proliferative tumors are often associated with improved therapeutic outcomes in chemotherapy and radiotherapy treatments[3–5]. However, despite the success of proliferative signatures as predictors of treatment response, there are unfortunately still many patients with high-proliferative tumors that exhibit high levels of resistance to treatment[6–9]. The mechanisms that underlie this paradoxical resistance remain largely unknown.

Chemosensitivity in tumors is the result of heterogenous concomitant proliferation-dependent and proliferation-independent resistance mechanisms[10,11]. To understand the role of heterogeneity in proliferation and resistance within a uniform genetic background, recent works have analyzed the single-cell behavior within genetically identical cell populations. These studies have revealed a widespread heterogeneity present in cell proliferation with subpopulations of cells following very diverse proliferation patterns. When cells are clustered by the total number of divisions, their proliferative behavior is remarkably similar within different clones and across cell line models from very different tissues, such as bone, retina, lung, and breast, suggesting a common mechanism behind it[12–15]. To test how different proliferation behaviors affect sensitivity, in a previous study we tracked hundreds of individual cells before and after chemotherapy and showed that, contrary to expectation, high proliferative cells were less susceptible to cell death[12]. Importantly, the underlying mechanism driving the heterogeneous proliferation patterns and their differential sensitivity remains mostly unknown.

Upon DNA damage induction, histone H2AX is phosphorylated (γH2AX) by Ataxia-telangiectasia mutated (ATM) initiating the recruitment of DNA damage response proteins. ATM activates the transcriptional activity of p53 at its targets, such as the cyclin-dependent kinase inhibitor p21 and the E3 ubiquitin ligase Mdm2, a negative regulator of p53[16,17]. This leads to p53 accumulation and subsequent generation of p53 pulses through the p53-MDM2 negative feedback loop. Single cell studies have demonstrated that incremental levels of radiation-induced DNA damage increase the fraction of pulsating cells while keeping the amplitude and width of p53 pulses relatively stable[18–20]. While the mechanisms of p53 and p21 activation upon exogenous DNA damage are qualitatively well understood, it remains unclear how this network quantitatively encodes endogenous DNA damage levels and its role in the heterogeneous proliferation patterns.

Here, we investigate at the single-cell level the interplay between DNA damage, the long-term activity of the p53-p21 signaling, and the individual proliferation activity of previously published datasets[12,14]. Our aim is to understand how these three factors quantitatively relate to each other, and how they might contribute to maintaining proliferation heterogeneity. We identified a gradual scaling law between DNA damage, cell proliferation, and the activity of p53 and p21. Our analysis showed unexpected changes in the p53 and p21 amplitude and the inter-pulse p53 period that encode damage strength and changes in proliferation. Furthermore, we identified a temporal switch in p53 oscillatory properties that allows a subpopulation of cells to

escape from a low proliferation to a high proliferation state, highlighting the complex and dynamic interplay between these factors.

## Results

**Endogenous DNA damage tunes proliferation activity**. The speed at which a population of cells grows is determined by how quickly individual cells progress through their cell cycle, the transition rate at which they enter or exit a quiescence state, and the rate of cell death[21]. External and internal stress factors such as exogenous and endogenous DNA damage as well as cellular factors like genetic and epigenetic modifications contribute to affecting these transition rates and consequently the outcome growth patterns. Consequently, the heterogeneity underlying the growth patterns' mechanisms arises from different individual proliferation behaviors.

To study how individual cells' proliferation activity relates to DNA damage levels we built up and expanded the analysis of a single cell dataset from our group, originally published in[12]. The experimental design combines a live recording with immuno-fluorescent imaging of the same cells. For the analysis, hundreds of individual cells were tracked for 52 h annotating the time of each division event, and, at the end of the live recording, cells were fixed and stained with a nuclear and DNA damage reporter (Fig. 1a). The division profiles revealed very heterogeneous proliferation behavior: one group of low proliferative cells (arrested or dividing only once) and a main group of high proliferative cells (2 to 3 divisions) (Fig. 1b). Note that these two proliferative groups maintain their proliferative differences for multiple days (see Fig. 1Sa, where we used the first two days to quantify their proliferation identity and three days to evaluate the posterior behavior). Next, we assessed the extent of DNA damage in all monitored cells by measuring the average nuclear γH2AX signal, a canonical marker for DNA double-strand breaks (Fig. 1c)[22]. Our findings reveal that the majority of these cells exhibit low mean nuclear GFP-γH2AX levels (as shown in Fig. 1d). Moreover, we relate the division profile of each live recorded cell with their corresponding GFP-γH2AX values (Fig. 1e). This analysis shows that cells undergoing a higher total number of divisions exhibit lower levels of DNA damage by the end of the recording period (see Fig. 1f and Table 1 and also Fig. 1Sb for alternative violin plots). We then asked if this correlation resulted from a DNA damage-induced prolongation of the time in between cell division events, referred to as the intermitotic time, and the limited observational time of our experiment. For this, we determined for every dividing cell the intermitotic time and their level of DNA damage, which resulted in a poor correlation with the cells' DNA damage (Fig. 1g).

In Fig. 2 we explore potential scenarios behind the observed poor correlation and the relationship with population growth. Exponentially growing cell populations might result from heterogeneous individual proliferation patterns (Fig. 2a). Cells can proliferate at different rates ranging from high activity (increased number of divisions) to low activity (reduced number of divisions). We hypothesize two alternative scenarios that could underlie such heterogeneous proliferation patterns. In the first scenario, the number of divisions relates directly to the intermitotic time of cells, so cells that divide more slowly will experience fewer total divisions within a given time frame (Fig. 2b). In the second scenario, cells have similar intermitotic times, but the different number of division events (proliferation patterns) arises from cells entering quiescence at earlier or later time points.

Together this alludes that the levels of endogenous DNA damage might stop individual cells' dividing after a discrete

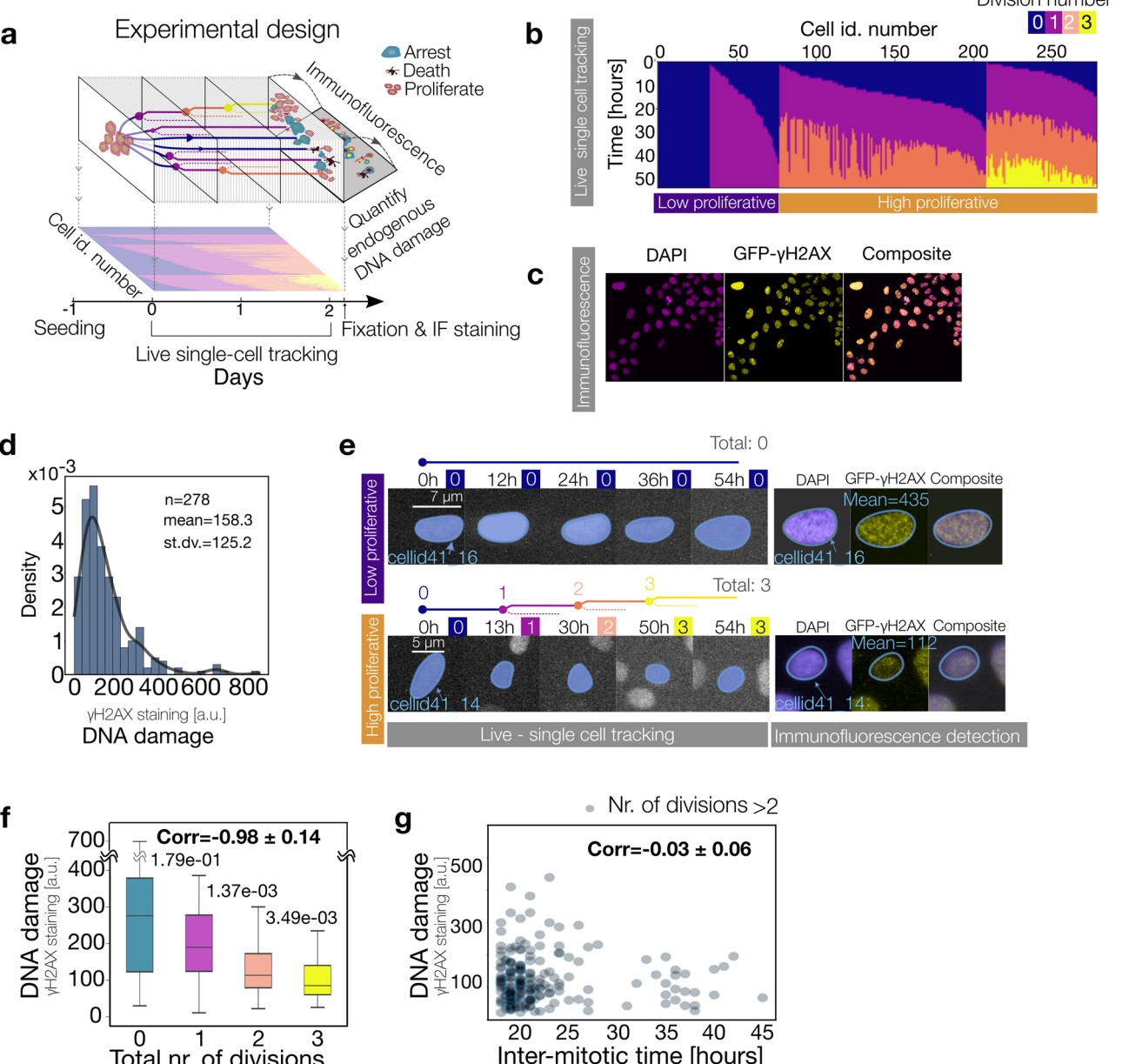

**Fig. 1 Endogenous DNA Damage tunes proliferation activity. a** Experimental setup for determining cellular proliferation patterns and the corresponding DNA damage levels in individual cells. Cells were seeded on day -1 and their proliferation behavior was tracked for 52 h. The top part of the scheme represents the heterogeneous proliferation patterns present in the population and their possible outcomes: arrest, death, and proliferation. The quantification of the division profile is projected in the bottom part. At the end of the recording cells were fixed and their levels of DNA damage were measured, linking live metrics of proliferation with DNA damage. **b** Individual division profiles were obtained after tracking cells for 52 h and annotating their division events. Each column represents the division activity of a single cell, with each mitotic event marked by a color transition (color code top right). Cells in each panel are clustered by their total number of divisions and then sorted by their time of first mitosis. **c** Immunofluorescence detection of DNA damage repair. Nuclei were identified by DAPI staining (left) and DNA damage repair activity by the phosphorylated H2AX (γH2AX—center) protein. The right image shows the composite DAPI and GFP-γH2AX staining. **d** The distribution of γH2AX levels of the recorded cells ($n = 278$) with the corresponding mean and standard deviation values in the upper right corner. **e** Snapshots of two representative cells with low and high proliferative behavior and their DNA damage levels. A high proliferative cell undergoes 3 divisions within ~2 days (bottom row) while a low proliferative cell remains with 0 divisions (top row). Top time labels show the timing of division events, and box numbers indicate recorded division events until that time point. Live single-cell tracking images show cells with a nuclear fluorescent signal for single-cell tracking (nucleus of tracked cell marked in blue). Immunofluorescence images show DAPI, GFP-γH2AX, and the composite of both channels at the end of the recording. **f** Boxplot of the DNA damage levels clustered according to the total number of divisions vs. the mean DNA damage levels of the cells with the corresponding p-value. The correlation coefficient is shown in the upper right corner. In the case of zero divisions, there were 33 recorded events, while for a single division, there were 44 events. In instances of two divisions, the count was 131, and for three divisions, there were 70 events. **g** Scatter plot of mean DNA damage levels of the cells versus their corresponding mean intermitotic time for each cell. The correlation coefficient is shown in the upper right corner.

number of divisions, proportional to the levels of DNA damage, rather than merely slowing down cell cycle progression (see Fig. 2c, d). Furthermore, we calculated the intermitotic durations for each proliferation subgroup categorized based on the total count of division events, and our findings indicated a sustained constancy in these intermitotic durations (see Fig. 2e untreated U2OS cells). Nevertheless, these results do not provide tools to discern a spurious from a causal correlation nor the causal direction of the relationship between proliferation and DNA damage. If DNA damage levels gradually drive proliferation differences, then the intermediate signaling pathways that control proliferation might quantitatively reflect this relationship.

**Gradual proliferation changes tuned with p53 and p21 amplitude**. To determine how the observed relationship between endogenous DNA damage and the proliferation pattern of

| Table 1 *p*-values to Fig. 1f. | | | |
| --- | --- | --- | --- |
| #divisions | 1 | 2 | 3 |
| 0 | 0.17936659 | 0.00014363 | 433E−06 |
| 1 | 0 | 0.00137446 | 923E−06 |
| 2 | 0.00137446 | 0 | 0.00349499 |
| 3 | 923E−06 | 000349499 | 0 |

individual cells was reflected in the activity of intermediate DNA damage signaling factors, we focused on the response of p53 and p21, one of p53's main transcriptional targets involved in cell arrest. For this, we analyzed one of the longest published datasets of individual human RPE (retinal pigment epithelium) cells treated with a gradient of radiation-induced DNA damage see Fig. 3a and ref. [14]. Cells were exposed to 0, 2, 4, and 10 Gy of gamma radiation at time $t = 0$ and their proliferation, p53, and p21 activities were tracked at the single cell level for over 5 days (Fig. 3a, b and additional example traces in 2Sa). First, we examined the overall impact of the incremental radiation doses and confirmed an increase in the arrested fractions with 5%, 42%, 67%, and 98% respectively, and no cell death events (Fig. 2Sb). To determine how differences in proliferation outcomes emerge over time, we quantified the cumulative distribution of division events for each treated group (Fig. 3c). The results revealed an evolution of dose-dependent population responses throughout the whole recording. Population growth changes could result from inter-mitotic time prolongations driven by the initial DNA damage. To evaluate this possibility, we calculated the mean intermitotic time within each dose and observed that it remained relatively constant across all conditions (see Fig. 2f RPE cells). Altogether, these results suggest that contrary to the expectation, the population growth effects of the radiation dose are not simply driven by intermitotic time prolongations. To differentiate between scenarios involving intermitotic time prolongation (Fig. 2d scenario

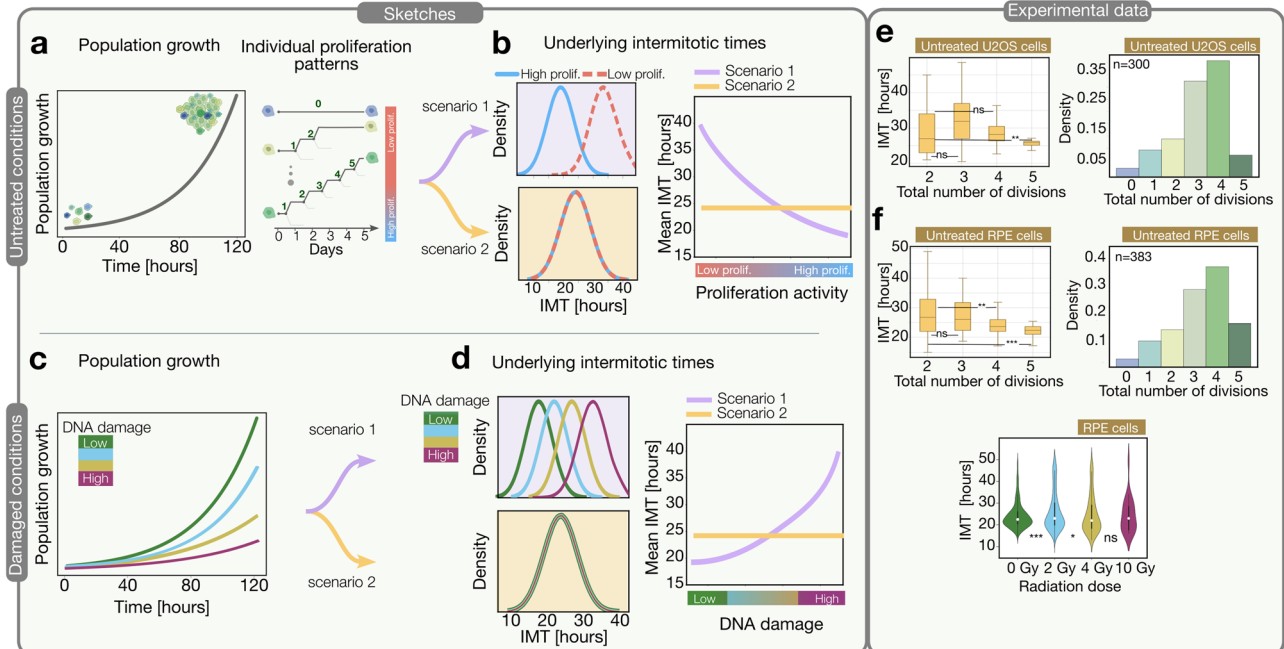

**Fig. 2 Population growth emerges from heterogeneous proliferative patterns of individual cells. Possible scenarios of how proliferation patterns of individual cells can lead to population growth. a** Untreated growth conditions. Sketch of cells' population growth (left) and individual proliferation patterns sorted by total number of divisions from low to high proliferation activity (right). **b** Sketch of two alternative proliferation scenarios. Scenario 1 (purple): low to high proliferation groups have distinctive distributions of intermitotic time (IMT), whereas, in scenario 2 (yellow), both groups share the same IMT distribution (left). Correspondingly, mean IMT decreases as mean proliferation activity increases (scenario 1) or remains constant (scenario 2). **c** DNA damaged conditions: Sketch to explain how elevated radiation exposure causing DNA damage can decrease cell population growth. **d** Sketch showing how the population growth reduction can result from two scenarios: DNA damage prolongs the IMTs in a dose-dependent manner (scenario 1: purple). IMTs remain stable, but cells might enter quiescence earlier in a dose-dependent manner, reducing overall growth rates (scenario 2: yellow). **e** U2OS experimental data: The left plot shows the intermitotic times of each group classified by the total number of divisions as a box plot with the corresponding p-values. The right plot shows the distribution of the total number of divisions. Sample sizes for untreated U2OS cells were $n = 37$ (2 divisions), 94 (3 divisions), 114 (4 divisions), and 21 (5 divisions). **f** RPE experimental data: The left box plot represents the times for each subgroup, which are categorized based on the total number of divisions with the corresponding p-values. On the right, we have the distribution of the total number of divisions. At the bottom, the violin plot represents the intermitotic times of RPE cells under varying radiation conditions. For the upper plots, sample sizes are $n = 44$ (2 divisions), 86 (3 divisions), 170 (4 divisions), and 50 (5 divisions).

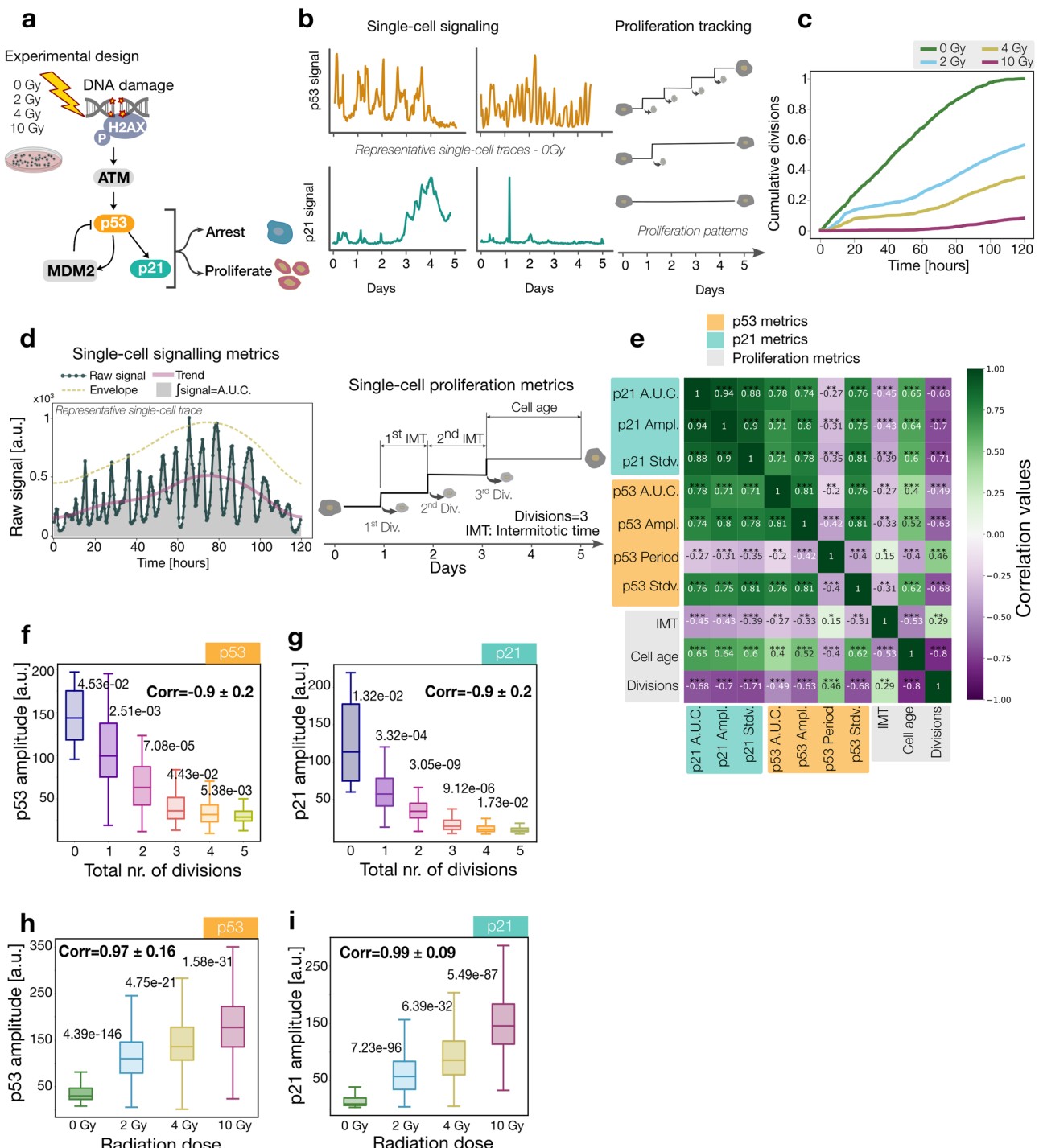

1) and DNA damage-level-dependent quiescence entry times (Fig. 2d scenario 2), further experiments utilizing reporter cell lines capable of detecting quiescence entry are necessary[14,23,24].

Previous studies have shown that the activity of the p53 signaling network is capable of encoding the type as well as the strength of the DNA damage. Short-term recordings indicated that DNA damage doses are encoded through the total number of p53 pulses without affecting the pulses' amplitude or duration[18–20]. To evaluate how the long-term signaling features of the p53 network quantitatively relate to proliferation and DNA damage doses, we parametrized a set of canonical time-series signal metrics and three single cell proliferation metrics: intermitotic time, cell age, and the total number of divisions

(see Fig. 3d and Methods). To confirm the amplitude derived from the continuous wavelet transform, we conducted a comparison with values computed through the pick-peaking method, revealing a notable equivalence (for additional information, refer to the Methods section and Fig. 2Sc). To analyze the relationship between signaling and proliferation, we quantified correlations among all metrics for each radiation dose (0 Gy in Figs. 3e and 2, 4, and 10 Gy in Fig. 2Sd). These analyses revealed that the p53 amplitude and p21 amplitude were the most correlated with the proliferation metrics from all properties. In comparison, the number of p53 pulses (calculated through peak-picking) and the period of p53 pulses (calculated using continuous wavelet transform) showed only poor correlations.

**Fig. 3 Gradual proliferation changes tuned with p53 and p21 amplitude. a** Workflow describing the experimental setup of the analyzed dataset. RPE cells were damaged with different radiation strengths at $t = 0$. The signaling pathway responsible for the cell response to DNA damage is shown together with the corresponding outcome. **b** p53 and p21 activity together with single-cell division events were recorded for 5 days. **c** Cumulative distribution of division events for cells within each radiation dose (see Methods). **d** Representative single-cell p53 raw signal (dark green line with dots) showing different components analyzed (top panel): amplitude envelope (yellow dashed line), trend (pink continuous line), and area under the curve (A.U.C.) in gray. The bottom panel illustrates the intermitotic time and age of a representative cell. **e** Heatmap of the mean single-cell correlation coefficients of p53 (yellow), p21 (green), and proliferation metrics (gray) for the untreated cells. A positive high correlation is shown in green and a negative correlation in purple. The correlation coefficients are shown in the respective boxes (see Methods). **f** Boxplot of the median p53 amplitude of cells in each condition using the continuous wavelet transform (see Methods), with the corresponding *p*-values between consecutive groups and the correlation coefficient. The sample size is the following: $n = 9$ (0 divisions), $n = 24$ (1 division), $n = 44$ (2 divisions), $n = 86$ (3 divisions), $n = 170$ (4 divisions), and $n = 50$ (5 divisions). **g** Boxplot of the median p21 amplitude of cells in each condition using the continuous wavelet transform, with the corresponding *p*-values between consecutive groups and the correlation coefficient. The sample size is the same as for (**f**). **h** Boxplot of median p53 amplitude for cells grouped by the received radiation, with the corresponding p-values between consecutive groups and the correlation coefficient. In each group, we had $n = 383, 892, 739$, and $842$, respectively. **i** Boxplot of median p21 amplitude for cells grouped by the damaging dose, with the corresponding *p*-values between consecutive groups and the correlation coefficient. The same number of cells as in (**h**) was considered for this plot.

**Table 2 *p*-values to Fig. 3f.**

| #divisions | 1 | 2 | 3 | 4 | 5 |
|---|---|---|---|---|---|
| 0 | 4500E−02 | 1030E−04 | 2520E−05 | 2030E−05 | 1310E−05 |
| 1 | 0000E+00 | 2500E−03 | 1200E−05 | 4050E−06 | 1230E−06 |
| 2 | 2500E−03 | 0000E+00 | 7080E−05 | 1330E−06 | 2760E−08 |
| 3 | 1200E−05 | 7080E−05 | 0000E+00 | 4400E−02 | 6670E−05 |
| 4 | 4050E−06 | 1330E−06 | 4400E−02 | 0000E+00 | 5400E−03 |
| 5 | 1230E−06 | 2760E−08 | 6670E−05 | 5400E−03 | 0000E+00 |

**Table 3 *p*-values to Fig. 3g.**

| #divisions | 1 | 2 | 3 | 4 | 5 |
|---|---|---|---|---|---|
| 0 | 1300E−02 | 1500E−03 | 5100E−04 | 3800E−04 | 3370E−04 |
| 1 | 0000E+00 | 3321E−04 | 5827E−07 | 1051E−07 | 5803E−08 |
| 2 | 3321E−04 | 0000E+00 | 3049E−09 | 7331E−13 | 5533E−14 |
| 3 | 5827E−07 | 3049E−09 | 0000E+00 | 9116E−06 | 7284E−09 |
| 4 | 1051E−07 | 7330E−13 | 9116E−06 | 0,000E+00 | 1727E−02 |
| 5 | 5803E−08 | 5533E−14 | 7284E−09 | 1727E−02 | 0000E+00 |

We next explored how the most correlated metrics, i.e. p53 and p21 amplitude, varied for untreated cells clustered by their total number of divisions (Fig. 3f, g and Tables 2 and 3, alternative violin plots can be found in Fig. 2Se). This bulk analysis showed that as proliferating activity increases, cells progressively reduce their p53 and p21 levels, in agreement with other recent studies[25]. To assess if the number of pulsatile cells differed substantially among doses, we calculated the fraction of pulsatile cells for each condition and observed that 68, 99, 99 and 100% of cells displayed pulsatile behavior for 0, 2, 4, and 10 Gy, respectively (see Fig. 2Sf and Methods). Thus, the majority of cells pulse in all conditions, an amount that remains constant for all treated conditions. Taken together with Fig. 1 these results suggest a scenario for untreated cells where endogenous levels of DNA damage proportionally increase p53 amplitude and p21 expression and are in tune with gradual changes in proliferation (see also Fig. 2Sg and h for radiation conditions and Table 1S and 2S). Nonetheless, these correlative results do not provide causality information, and alternative scenarios, such as replication-induced DNA damage, which could also result in strong correlations between DNA damage signal activation and proliferation metrics[26]. To test the relationship between proliferation, p53–p21 activation, and DNA damage, but to also decouple potential proliferation-induced DNA damage effects, we analyzed the signaling effects of exogenous DNA damage. For this, we evaluated how p53 and p21 amplitude changed for cells

**Table 4 *p*-values to Fig. 3h.**

| #dose | 2 Gy | 4 Gy | 10 Gy |
|---|---|---|---|
| 0 Gy | 490E−146 | 6720E−196 | 7850E−293 |
| 2 Gy | 0000E+00 | 4750E−21 | 2590E−95 |
| 4 Gy | 4750E−21 | 0000E+00 | 1580E−31 |
| 10 Gy | 2590E−95 | 1580E−31 | 0000E+00 |

**Table 5 *p*-values to Fig. 3i.**

| 2 | 2 Gy | 4 Gy | 10 Gy |
|---|---|---|---|
| 0 Gy | 7230E−96 | 2330E−157 | 0000E+00 |
| 2 Gy | 0000E+00 | 6390E−32 | 9030E−201 |
| 4 Gy | 6390E−32 | 0000E+00 | 5490E−87 |
| 10 Gy | 9030E−201 | 5490E−87 | 0000E+00 |

treated with 0, 2, 4, and 10 Gy. Our analysis of long-term recordings unveiled unexpected modulation of p53 and p21 amplitude, gradually encoding radiation dose strengths (Fig. 3h, I and Tables 4, 5, alternative violin plots can be found in Fig. 2Si).

**P53 period drives populational growth rate transition.** The above results indicate how amplitude rather than the p53 period

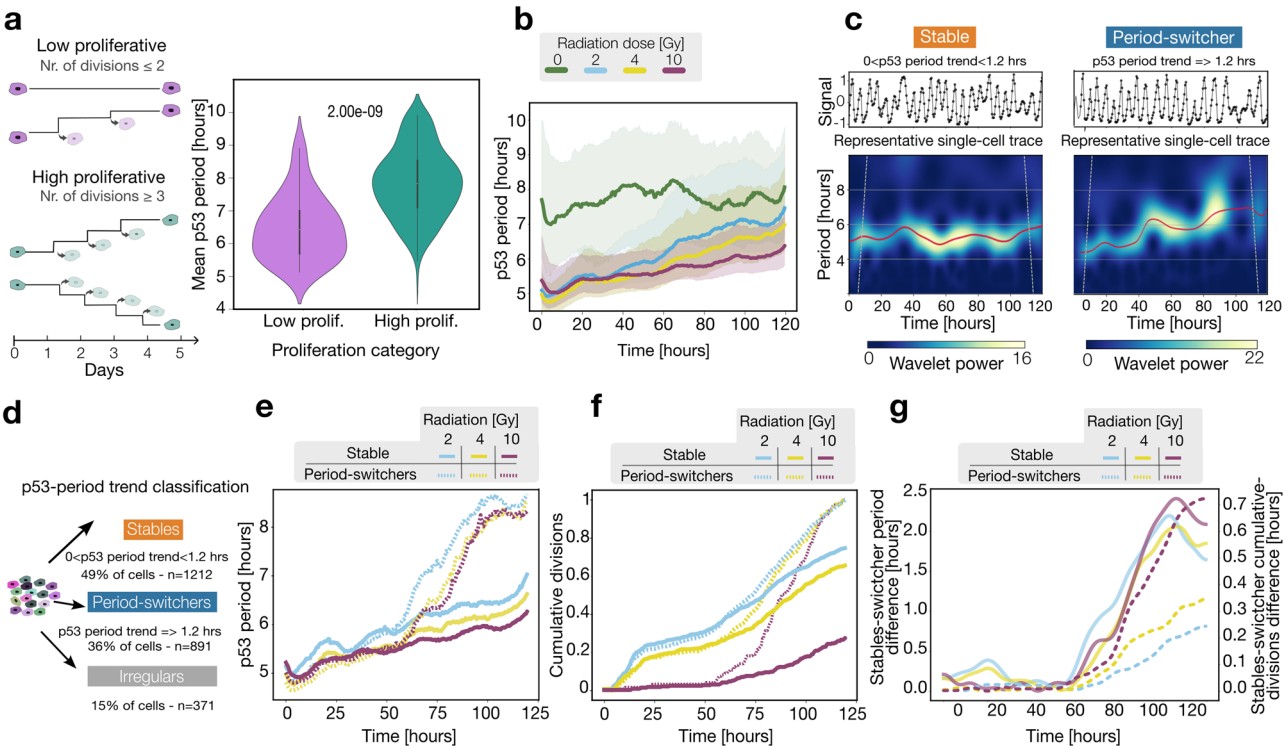

**Fig. 4 P53 period drives populational growth transition. a** Left: a sketch of the definition of low (up to 2 divisions) and high (more than 2 divisions) proliferative cells for a time frame of 5 days. Right: violin plots of mean p53 period for untreated cells (0 Gy) grouped corresponding to the low ($n = 33$) and high ($n = 350$) proliferative groups with the corresponding $p$-value. **b** Median instantaneous p53 period (dark line) of the population at each radiation condition with the corresponding quartiles (shaded area, see Methods). **c** Two representative p53 traces: left side with a stable period (stable) and right side with a prolonging period (period-switcher). The bottom panels show the corresponding wavelet power heatmap (see Methods). The dominant period, ridge frequency, is shown in red. The intervals of confidence (cone of confidence) are represented by the dashed gray lines. **d** Classification strategy for p53-period trends pooling all irradiated cells: 49% are 'stables' (<1.2 h trend over 5 days, orange), 36% are 'period-switchers' (>1.2 h trend over 5 days, blue), and 15% are 'irregulars' (unclassifiable, labeled in gray). **e** Median p53 period of each period-stability group of cells: cells with relatively stable p53 period (continuous line) and cells with increasing p53 period (dashed line). **f** Cumulative distribution of division events for cells that prolong their p53 period (dashed line) and cells that keep their p53 period relatively stable (continuous line). **g** The difference between the period of stables and period switchers is depicted with a continuous line (left axis) for each condition. The dashed line represents the difference between the cumulative distribution of division events of stables and period switchers (right axis).

best correlates with gradual proliferation differences within cells. Using a peak-finding algorithm, we noticed a gradual association between the number of pulses, and the radiation doses (Fig. 3Sa), suggesting that the damaging dose might have an influence on the p53 period. Given the previously established strong correlation between DNA damage levels and proliferation, we sought to determine further whether collective proliferation differences in untreated cells have a distinctive p53 period. To accomplish this, we classified cells into two groups (low and high proliferative) and calculated their mean p53 period (Fig. 4a). This classification indicated that highly proliferative cells pulse on average with a longer period than their lower proliferative counterpart ($p$-value = 2e−09). Consistent with previous works indicating that the number of p53 pulses encodes DNA damage (see Fig. 3Sa), our results suggest that low proliferative cells, potentially carrying higher endogenous levels of DNA damage, oscillate more frequently.

Time-averaged measurements, such as e.g. autocorrelation and Fourier transform, provide robust metrics especially suited for signals' properties that remain constant in time, so-called stationary signals. Testing stationarity in this dataset, using the robust Augmented Dickey-Fuller test revealed a proportion of cells with non-stationary p53 signals highlighting the need for time-dependent signal analysis (Fig. 3Sb). Thus, we analyzed the time evolution of signals' parameters, for cells within each

radiation dose. To minimize amplitude fluctuation effects that could result in period detection artifacts, for this analysis all signals are detrended and amplitude normalized (see Methods). Contrary to the observations in short-term recordings, our analysis revealed that radiated cells show an unexpected trajectory of p53 periods prolonging in a time- and dose-depended manner while untreated cells' period fluctuated around 7.5 h (Fig. 4b). To test whether the collective period prolongation of radiated cells resulted from a population-average effect of cells increasingly pulsing with longer circa 7.5 h period or from individual cells prolonging their period at different rates, we classified all radiated cells based on their period evolution (Fig. 4c). This resulted in three groups: cells with stable periods (49% of all damaged cells), period-switchers (36% of all damaged cells), and those who failed to be categorized (15% of all damaged cells) (Fig. 4d, 3Sc for a classification by dose). Computing the period evolution of cells classified as stable or period-switchers for each radiation group indicated a time window between 50 and 75 h for the period transition (Fig. 4e). Since the period of switcher cells, within each dose group, approached progressively the characteristic period of untreated cells, we hypothesized that the period transition of individual cells could be accompanied by a corresponding increase in the proliferation pattern. To test this hypothesis, we computed the cumulative division events for cells clustered as in Fig. 4d and found that cells that maintained a stable period

**Table 6 slope to Fig. 4f.**

|           | 2 Gy   | 4 Gy   | 10 Gy  |
|-----------|--------|--------|--------|
| Stables   | 0.0065 | 0.0072 | 0.0041 |
| Switchers | 0.0109 | 0.0133 | 0.0178 |

proliferated less, while cells that prolonged their period diverged into a higher proliferative group (Fig. 4f). The differences in proliferation activity between groups are synchronized with the period divergence and became more pronounced as the radiation dose increased (Fig. 4g). Calculating proliferation rate differences between stables and switchers after the bifurcation time of around 60 h indicates a dose-dependent boost effect with a 1.68, 1.84, and 4.49 fold-increase in proliferation for 2, 4, and 10 Gy respectively (see rates estimations in Table 6 and Methods Section). It is important to note that these results represent variations in the rate of division events between the stable and period-switchers subpopulations within each dose and do not reflect the relative effect between doses nor indicate the proportion of cells entering cell cycle arrest. Moreover, randomly clustering cells in similar-sized groups resulted in no significant period nor proliferation differences (Fig. 3Sd). Our findings suggest that the oscillatory rate of p53 modulates cell growth programs. Further research is needed to elucidate the specific mechanisms by which the p53 period switch regulates growth.

## Discussion

In this study, we quantitatively characterize the functional relationship between DNA damage levels, key signaling players of the DNA damage response, and the proliferation behavior of individual cells. We found that in a population of cells DNA damage levels are quantitively encoded in the amplitude of p53 and p21 regulating proliferation in a gradual manner. Our results suggest a model where the p53 network is capable of sensing different levels of DNA damage and gradually fine-tuning proliferation at the populational level. A direct consequence of such a model is that a population of cells with a heterogeneous distribution of endogenous DNA damage results in a heterogeneous distribution of proliferation patterns, as is observed in cells from very diverse tissues such as the retina, breast, lung, and bone. This positions endogenous DNA damage as a fundamental source of proliferation heterogeneity, with important potential implications for the growth and homeostasis of tissues, as well as for the formation of tumors. Moreover, by studying the relationship between endogenous DNA damage, p53 activation, and proliferative signatures, our work provides a framework where individual proliferation signatures can be disentangled from non-genetic priming differences, and so help understand the paradoxical cases where highly proliferative cells exhibit resistance to treatment[8,9]. Our findings showing strong relationships between DNA damage, signaling, and proliferation do not offer means to determine causality, nor do they establish the causal direction of the relationship between proliferation and DNA damage. Further research is needed to validate this model and, more importantly, to understand the mechanisms driving the heterogeneity of endogenous DNA damage while establishing a connection to p53 dynamics and cell proliferation at the single-cell level.

The dynamics of signaling molecules in mammalian cells are capable of encoding features of stimuli as well as activating downstream response programs[27–29]. Previous studies have identified global changes in the p53 signaling dynamics, e.g. pulsatile versus sustained, as the determinants of the individual cell responses upon genotoxic stress[28,30,31]. Here we identified quantitative changes of a specific parameter of p53 signaling, i.e.

the p53 inter-pulse period, as a regulator of growth rate changes. Around 60 h post DNA damage, a sudden prolongation in the period of p53 boosts proliferation in a subpopulation of cells, e.g. a 4.5-fold proliferation boost for 10 Gy radiated cells (Fig. 4e–g). This suggests novel perspectives on the fundamental mechanisms of proliferation and emphasizes a new function of p53 in governing cell growth. Thus, complementing previous studies on qualitative differences in signaling dynamics our work highlights the importance of considering quantitative changes over an extended period to understand the emergence of resistance subpopulations. Faster MDM2-mediated degradation of p53 has been recently indicated as the main regulator of the p53 period[32,33]. Identifying specific MDM2 and p53 mutations that affect the signaling dynamics or pharmacologically targeting the p53 period might provide new therapeutic avenues to control the re-growth of radiated cells. Additional investigation is essential to comprehend the mechanism by which the p53 period controls downstream programs that affect cell growth and the potential implications for new therapeutic targets.

In this study, we draw conclusions integrating observations from a cancer and a non-cancer cell line model. Future research within a single cell line and across various cell lines will clarify whether the observed correlations between DNA damage, proliferation, and p53/p21 is a general trend or if different relationships govern the responses to DNA damage in other cell lines. Furthermore, our assessment of intermitotic times is constrained by the duration of our recordings, which could introduce a bias favoring shorter IMT, potentially obscuring the genuine cell cycle durations. To address this concern, extending recording durations and implementing statistical models that consider various influencing factors could yield more accurate estimations of the underlying cell cycle lengths.[34,35]

Proliferation heterogeneity is ubiquitous in mammalian cells and its regulation and functional significance remains unclear. Predicting such fundamental behavior as individual cell proliferation behavior, remains a complex fundamental problem in biology, even in well-controlled constant in-vitro conditions. Our work highlights the importance of studying long-term cell responses at the single-cell level through continuous recordings of cell growth and signaling dynamics. To fully capture the features of such complex signals and determine their functional role, a systematic quantification of non-stationary time-series analysis metrics is essential. Such recording and analysis approaches may accelerate the identification of specific signal parameters to understand how molecules dynamically encode information in the context of DNA damage response.

## Methods

**Histogram plots**. Histogram plots were computed using the Python function `histplot` from `seaborn` package[36].

**Correlation coefficients**. Correlation coefficients were calculated using the Python function `corrcoef` from `Numpy`[37] and `pearsonr` from `SciPy`[38], while accounting for the medians of p53 and p21 dynamics, or the levels of DNA damage, within distinct groups characterized by varying total numbers of divisions. For the error calculation, we used the standard error formula.

**Cumulative distribution of division events**. To calculate the cumulative distributions (CMs), we implemented the Python script `cumdistr_all_divisions.py`. In this script, we condensed all divisions across all cells into a single temporal vector and subsequently applied the `cumsum` function from Python to do an accumulative sum of the division events. To

mitigate potential sample size effects among different radiation doses, a reference number of cells was randomly selected from the condition with the smallest sample size (383 cells for the 0 Gy condition) and used to randomly select the same number of cells from larger sample size conditions. This process was repeated iteratively 10 times to obtain the average cumulative distributions presented in the main text, Fig. 3c. Finally, we normalized all the conditions to the maximum of the cumulative distribution of the untreated condition.

**Violin plots**. Violon plots were calculated using the Python function `violinplot` from the `seaborn` package[36] with the following parameters `showfliers = False` and `cut=0`.

**Box plots**. Box plots were calculated using the Python function `boxplot` from the `pandas` package[39] with the parameter `showfliers = False`. The correlation coefficients were calculated with the function `corrcoef` from `Numpy`[37] of the median values of each group, as explained in the point "Correlation coefficients" from this section.

**Detrending**. Raw data were detrended using `pyBOAT` software package for Python (v0.9.2) function `sinc_detrend` using a cut-off period of 50 h[40], in particular `pyBOAT` uses a sinc filter that removes the periods larger than a certain cut-off.

***P*-values**. *p*-values were computed using the Python function `ttest_ind` with Welch's *t* test to compare two datasets with different sample sizes from the package `stats` of `Scipy`[38].

**Continuous p53 and p21 amplitude calculation**. Continuous amplitude envelope calculation was obtained using continuous wavelet transform implemented in the open-source software package `pyBOAT` through the functions `compute_spectrum`, `get_maxRidge`, and `ridge_data`. We computed the continuous wavelet transform, implemented in `pyBOAT`, to identify the predominant oscillatory elements characterized by the highest power through ridge detection for individual signals. Then, from the wavelet power spectrum, we estimated the amplitude spectrum using the Morlet Wavelet scaling factor[40]. To consider the consistency in amplitude derived from pyBOAT refer to Fig. 2Sc, where we contrasted amplitude values obtained using two different methods: continuous wavelet transform (`pyBOAT`) and peak picking (`Scipy`). For the amplitude computation from `pyBOAT`, we considered detrended signals where the trend with periods larger than 50 h was subtracted. For each cell, we obtained the median amplitude from the ridge detection after applying the continuous wavelet transform as explained above. Conversely, we computed the peak heights using the Python function `find_peaks` from `Scipy` with the following parameters (`height=50, distance=7, prominence=50`). For comparison, a scatter plot of the median values for the amplitude obtained with the different methods for each condition is presented, along with the corresponding correlation coefficient computed using the Pearson correlation.

**Correlation coefficients heatmap**. Pairwise correlation coefficients for several metrics were calculated for each individual cell using the `corr` function from the `pandas` package of Python. We calculated the intermitotic intervals by measuring the time between successive division events and subsequently determined the median values for each cell. The calculation of the area under the curve (AUC) involved determining the mean value resulting from two distinct methods, namely the trapezoidal and Simpson methods. As previously described, we obtained the amplitude as a continuous curve from `pyBOAT` (a vector with the heights of the peaks). The standard deviation, on the other hand, was computed based on the detrended signal obtained from `pyBOAT` as well. For each continuous metric, we extracted the median value per cell and subsequently conducted calculations to determine the correlation coefficients across all individual cells.

**Bar plot**. The bar plot was computed using the Python function `barplot` from `seaborn`[36]. To compute the fraction of arrested cells, we calculated how many cells divided zero or one time during the first 48 h of the experiment.

**Number of pulsatile cells**. The number of pulsatile cells was computed with the Python function `find_peaks` from `Scipy` with the following parameters (`height=50, distance=7, prominence=50`). A cell was considered pulsatile when it exhibited at least 10 pulses in 120 h (the recording duration).

**Instantaneous period calculations**. Instantaneous periods from individual cells are calculated from detrended raw data using the functions `compute_spectrum`, `get_maxRidge`, and `ridge_data` from `pyBOAT` with a cut-off of 50 h and an amplitude normalization with a time window of 50 h. The amplitude normalization was done by taking the inverse of the envelope of the detrended signal.

**Mean period calculations**. In detrended p53 raw data, the period of individual p53 traces is calculated using the `pyBOAT` function `compute_spectrum`, `get_maxRidge`, and `ridge_data` to obtain instantaneous values, and then with the `NumPy` function `mean` we get the mean values. Instantaneous periods and then time-averaged for each individual cell and then all individual cell periods are plotted.

**P53 period stability classification**. To estimate the period stability over time, the instantaneous period of the main signal component was calculated using `ridge analysis` (from `pyBOAT`), and a linear fit was implemented to estimate the slope of the instantaneous period trend. A slope threshold of 0.01 [1/h] was used to classify cells as period stable or period switchers. Slopes between 0 and 0.01 [1/h] are classified as stable whereas slopes above 0.01 [1/h] are classified as period switchers. Using a 5-h period signal as a reference, cells were further classified as stable (period changes less than 5% within a cycle), prolongers (higher period changes), or irregular (period changes outside these two categories).

**Difference between period/cumulative distribution of division events of stables and period-switchers**. The period of the stables was subtracted from the period-switchers and smoothed using the `sinc_smooth` function of `pyBOAT` with a cut-off period of 10 h.

**Proliferation slope**. The proliferation rate of the stables or period-switchers population was estimated through a linear fitting of the increasing part using the Python function `fit` from `NumPy` (see Table 6).

**Peak picking of the pulse number**. The number of peaks per cell within a particular condition was computed using the Python function `find_peaks` from `SciPy`. A signal was passed to this function with the following parameters: (`height=50, distance=8, prominence=50`). The cells that had more than

two peaks were collected and plotted using the `violinplot` function from `seaborn`.

**Augmented fully Dickey test**. To compute how many signals were non-stationary we used the Python function `adfuller` from `statsmodels`[41] and classified as non-stationary those that had a p-value higher than 0.05.

**Random p53 period classification**. Within a certain radiation condition, 300 cells were chosen randomly for each group using the Python function `sample` from the `random` package. This process was iterated 20 times to minimize the bias between iterations (see Fig. 3Sd).

## Data availability
All data supporting the findings of this investigation are available in the previously published papers[12, 14].

## Code availability
The code implemented for the above analyses can be found here: https://github.com/Granada-Lab/proliferation-p53-p21 or from the authors upon request.

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

## Acknowledgements
The authors would like to express our gratitude to the Galit Lahav laboratory and José Reyes for sharing the data. We are also grateful to Manuela Benary and the members of the Ulrich Keilholz laboratory for their assistance and relevant feedback throughout the project. We would like to acknowledge the inputs provided by Michela di Virgilio and her laboratory. Finally, we acknowledge the financial support provided by the German Federal Ministry for Education and Research (BMBF) through the e:Med Juniorverbund DeepLTNBC TP 3 - 01ZX1917C that enabled us to carry out this research.

## Author contributions
Conceptualization: N.G., U.K., H.H., A.E.G. Methodology: N.G., A.E.G, Data analysis: N.G., N.B. Investigation: N.G., A.E.G. Intellectual support: U.K., H.H. Visualization: N.G., A.E.G. Supervision: U.K., H.H., and A.E.G. Writing—original draft: N.G., A.E.G.

## Funding

## Competing interests

The authors declare no competing interests.
