## [Peer Review File · Communications Biology]

Reviewers' comments:

Reviewer #1 (Remarks to the Author):

This study extends from a prior work conducted by the corresponding author (PMID 32049575), which revealed that the predictive outcome of chemotherapy-induced change in cell proliferation is linked to the proliferative history of cells, rather than their cell cycle status. In this current study, the authors further investigated the underlying mechanism for heterogenous proliferation patterns (division event counts and inter-mitotic time) across both basal and genotoxic conditions. They incorporated experimental data from another study (PMID 30057196) to demonstrate that the oscillatory amplitude of p53 can encode information for p21-mediated cell cycle arrest. The main findings of this study are as follows: 1. Cells with low proliferative and high proliferative history exhibit different numbers of division events, but possess similar inter-mitotic time, suggesting that cells differ in their tendency entering quiescence; 2. Levels of DNA damage, p53 and p21 amplitudes show an inverse correlation with the number of division events; and 3. the shift of p53 period accompanies the escape from arrest in low proliferative cells. In summary, the study provides insights into the potential mechanistic link between cell proliferation and signaling dynamics. However, it is important to note that the authors' observations are interesting but preliminary. Rigorous data analyses, precise data interpretation, and the direct experimental evidence supporting their hypothesis are strongly recommended before considering it for publication.

Major concerns:

1. Drawing the conclusion that "DNA damage levels are quantitatively translated into p53 and p21 signal parameters, resembling an "analog" system rather than a "digital" one" could be problematic and misleading, as this conclusion mainly relies on correlations of single-cell outcomes related to DNA damage, p53 amplitude, and proliferation patterns within cell populations. It is important to note that characterizing a cell population as an "analog" system doesn't exclude the possibility of heterogenous "digital" systems present within individual cells. Moreover, it is more challenging to reveal "digital" responses using correlation analysis, especially when the variation is high (see the error bars in Figure 1G & Figure 2F-I). To claim an analog response, the authors are recommended to show the distribution of single-cell responses in their Figure 1G & Figure 2F-I (e.g., using violin plots, instead of box plots) to demonstrate graded responses in cell populations, rather than bifurcations.
2. Conventionally, the p53 oscillator is perceived as an excitable system that maintains its amplitude regardless of the intensity of DNA damage (PMID: 21556066). However, the authors showed a dose-dependent p53 amplitude (Figure 2F & 2H). How does this phenomenon arise? Besides from that, the authors demonstrate a measurable p53 amplitude even at zero Gy of irradiation, a scenario where one might anticipate the absence of p53 oscillations. Is the measured p53 amplitude here indicative of the post-mitotic pulse (PMID 20598361) or merely noise within the dataset? The authors should provide example traces to clarify this point.
3. The authors showed the inverse correlation between DNA damage and the number of cell divisions in U2OS (figure 1G), while presenting the results of p53/p21 in RPE cells (figure 2F). The observed correlation established across two distinct cell lines, might not necessarily hold true within a single cell line or consistently across various cell lines. The authors need to show the correlation in the same cell line, ideally in two different cell lines, to demonstrate the correlation holds in the same cell line and generally occurs in different cell lines.
4. In their Box, the authors argued that "Cells have similar intermitotic time, but the different number of division events (proliferation patterns) arises from cells entering quiescence at earlier or later time points". To make this argument, the authors need to provide experimental evidence showing the entry of quiescence (e.g., by monitoring APC/C or CDK activity).

Minor concerns,

1. Data presentation needs to be improved. For example, it is not easy to comprehend the conceptual model (Figure 1A) and the experimental results (Figure 1G & 1H), as the current visual representation

does not align cohesively.

2. For most box plots, correlation coefficients are surprisingly high given how some boxes overlap quite strongly. Which data are these coefficients calculated from (the medians only or every datapoint)? What are the sample sizes? The authors should clarify those points and note the necessary information in the figure legend.

3. More details on analysis methods are needed. Merely referencing the Python scripts or functions used does not provide sufficient clarity for readers to assess the quality and robustness of the analysis procedures.

4. (Page 19) Figure 1A: "treshold" need to be corrected to "threshold".

5. (Page 19) Figure 1: The labeling for γ H2AX should be correct and consistent. The authors employed the label GFP-gH2AX in Figure 1D, while in Figure 1F they used γ H2AX. It is recommended to maintain uniformity in labeling throughout the study.

Reviewer #2 (Remarks to the Author):

The manuscript by Gutu et al presented new analysis of time-lapse data of DNA damaged cells and concluded that the p53-p21 network senses the distinctive DNA damage strength and then modulates cell proliferation. Although the data are interesting, I think more experimental data and data analysis, especially those that directly link DNA damage level, p53 dynamics and cell proliferation at the individual cell level, rather than population level, are needed in order to establish the functional correlation. It is also unclear to me what novel advance the study provides in terms of our understanding of cell fate control mediated by p53 upon DNA damage. Results from the paper appear to simply support the well-known scenario that higher DNA damage triggered higher upregulation of p53 and subsequently more cell cycle arrest (i.e., less cell division/proliferation). The authors need more single cell data to demonstrate the novel role of p53-p21 network stated in their study.

1. The most important data that this study needs are single cell time-lapse data that simultaneously measure the DNA damage level, p53 induction dynamics and cell fate in the same single cells. The authors should use cell lines that express both the GFP- γ H2AX reporter (shown in Fig. 1F) and a p53 fluorescence reporter (shown Fig. 2B), e.g., mRFP-p53, to acquire such data for quantitative analysis. All the correlation analysis of DNA damage level and p53 level in the current study are done at the population level. Given the large single cell variability, such population-level statistics that average over all single cells really miss the point of single cell analysis and may not reflect the genuine correlative behaviors in individual cells.

2. Moreover, the DNA damage level is obviously not at a steady-state level over time. As shown in Figure 1F (top panel), DNA damage is very high right after irradiation and then decreases over time, possibly due to DNA damage repair activated by p53. Does this decrease of DNA damage level alter p53 dynamics (e.g., switching) and cell fate compared to the early time points? Again this illustrates the importance of quantifying the dynamics of DNA damage level and p53 in the same cells over time, instead of simply using the average DNA damage strength as the metric to analyze the correlative behaviors.

3. The authors use the intermitotic time as a surrogate for cell cycle length. But in the intermitotic time calculation, the authors excluded the very long cell cycle because no division was observed. For example, in Figure 2D, from the 3rd division to the end of the imaging experiment, as there is no 4th division that occurred, the authors called the last phase "Cell age" and excluded it from the intermitotic time statistics. I think such analysis is wrong and artificially gave a shorter intermitotic time for scenario of high cell cycle arrest, e.g., under high DNA damage. In my opinion, if the cells do not divide, it can simply mean the intermitotic time is much longer than the duration of the imaging

experiment. Overall, I disagree with the authors' use of intermitotic time as a measure of cell cycle length, and I do not think their data make a distinction between prolonged cell cycle length and some sudden decision of no division.

4. After the cells divide, the DNA damage level, p53 and p21 level/dynamics may change in the daughter cells. The authors should quantify these features in individual cells before and after each cell division, and see how these features are genuinely correlated upon division. It is possible that cell division simply acts as a dilution scheme, i.e., DNA damage is lower and p53 is lower, so it further augments the proliferative potential. Again, I think much more single cell analysis have to be performed under distinctive scenario, instead of simply using the population or time averaged values to establish the correlative features.

5. On page 7 (lines 197-199), distribution of the cells showing different p53 dynamics is discussed. At what DNA damage level were these data obtained? What are the distributions of different p53 dynamics for untreated cells and cells at other DNA damage level, e.g., 2, 4, 10 Gy? It is stated in lines 209-212 that "Calculating proliferation rates differences between stables and switchers after the bifurcation time of around 60 hours indicates a dose-dependent boost effect with a 1.68, 1.84, and 4.49 fold-increase in proliferation for 2, 4, and 10 Gy, respectively". I am confused by this statement. As DNA damage level increases, more cells should get into cell cycle arrest so the proliferation at 10 Gy should be a lot lower than that at 2 and 4 Gy. Moreover, I would think at 10 Gy, most cells should be in a long cycle arrest and do not divide. What is the functional relevance and meaning of this boost in proliferation at the population level?

Minor points:

1. In Figure 1F, the labels "7 μM " and "5 μM " on the images should be "7 μm " and "5 μm "

2. Citation of "Box 1" is confusing. Can the author just rename it as a figure?

Reviewer #3 (Remarks to the Author):

This very interesting work by Gutu et al. explores the relation between DNA damage, the dynamic response of p21 and p53 activities, and proliferation patterns at the single cell level.

Through systematic analysis, the authors uncover a gradual relation between DNA damage, cell proliferation, and key features of p53 and p21 activity. They show changes in the amplitude of p53 and p21, as well as alterations in the inter pulse interval of p53, encoding damage strength and shifts in proliferation pattern. Finally, the analysis reveals a temporal switch in p53 oscillatory period, that correlates with a subpopulation of cells transitioning from low to high proliferation.

In my opinion this is an excellent work, a very rigorous study addressing relevant questions about how proliferation heterogeneity is produced in a population of cells. The data analysis is of very high quality and leads to novel conclusions and ideas that may be of broad interest. I have only a few comments for the authors to consider.

Comments.

[1] Fig. 1H: You conclude that there is a poor correlation between DNA damage levels and inter-mitotic times. Not sure I got this right, in this scatter plot you don't see cells with large DNA damage levels and long inter-mitotic times. Does the inverse of DNA damage levels correlate with inter-mitotic times?

[2] Box: this result is important. Should it be in a Figure then? I understand that it contains data relevant to both Figs. 1 and 2, so maybe you could split this.

[3] Fig. 2 and related text. The definition of amplitude was not clear to me. Is it the total amplitude of the signal, that is its absolute height? Or is it the amplitude from trough to peak values?

- Also, it is not clear what is correlated in Fig. 2E: In Fig. 2D you show a continuous amplitude envelope, as well as a continuous trend. These are curves, but then they are correlated to some scalar values like IMT (I assume this is the average of all IMTs of a given cell?). Is the average amplitude considered here? Reference to modules or functions from pyBOAT are given in the methods, but still I think that these things need some clarification in the text.

[4] From the abstract and introduction I got the impression that it was an experimental study. However, as far as I can understand, it relies on data analysis from previous work:

- Fig. 1 data: To evaluate how individual cells' proliferation activity relates to DNA damage levels we build up and expanded the analysis of a dataset from our group, originally published in the supplements of [12].

- Fig. 2 data: For this, we analyzed one of the longest published 125 datasets of individual human cells treated with a gradient of radiation-induced DNA damage, see (Figure 2A) and [14].

Still, in the Methods there is a section about U2OS cells. Are the experiments in the Box new? I think this should be stated more clearly to avoid wrong expectations.

Minor points.

[5] Fig. 1 A, maybe it would be more clear to plot the single cell behavior on the left for both types of population responses. At the moment it looks as if the left panel applies to both cases, but for the heterogeneous case the single cell behavior is plotted in the "Population" panels on the right.

- Fig. 1A, top right panel, I think it should be a solid line (like in the bottom panel) instead of dots, to make it clear it is the same thing, that is the population response.

- Fig. 1A, bottom right panel, I can barely see the colored response curves, can you make them thicker?

[6] Fig. 1F is not mentioned, it should come on page 4 together with 1G?

[7] Fig. 2A: I think RPE cells acronym is not defined.

[8] Fig. 2E: the text labels of correlations are tiny and cannot be resolved in normal size.

[9] page 6, line 178 - 180: "This classification indicated that highly proliferative cells pulse on average with a significantly lower period than their lower proliferative counterpart..." This should be ... slower period...?

[10] Augmented Dickey-Fuller test: the spelling is wrong (and mutating) in the text (line 187), in the Methods (line 344), and in Fig. S3B legend.

Reviewer #1 (Remarks to the Author):

This study extends from a prior work conducted by the corresponding author (PMID 32049575), which revealed that the predictive outcome of chemotherapy-induced change in cell proliferation is linked to the proliferative history of cells, rather than their cell cycle status. In this current study, the authors further investigated the underlying mechanism for heterogeneous proliferation patterns (division event counts and inter-mitotic time) across both basal and genotoxic conditions. They incorporated experimental data from another study (PMID 30057196) to demonstrate that the oscillatory amplitude of p53 can encode information for p21-mediated cell cycle arrest. The main findings of this study are as follows: 1. Cells with low proliferative and high proliferative history exhibit different numbers of division events, but possess similar inter-mitotic time, suggesting that cells differ in their tendency entering quiescence; 2. Levels of DNA damage, p53 and p21 amplitudes show an inverse correlation with the number of division events; and 3. The shift of p53 period accompanies the escape from arrest in low proliferative cells. In summary, the study provides insights into the potential mechanistic link between cell proliferation and signaling dynamics. However, it is important to note that the authors' observations are interesting but preliminary. Rigorous data analyses, precise data interpretation, and the direct experimental evidence supporting their hypothesis are strongly recommended before considering it for publication.

We express our gratitude to the reviewer for their time invested in reviewing our manuscript and the valuable remarks and feedback. In this response, we address each of the raised concerns, aiming to comprehensively resolve all the queries posed.

Major concerns:

1. (A) Drawing the conclusion that “DNA damage levels are quantitatively translated into p53 and p21 signal parameters, resembling an “analog” system rather than a “digital” one” could be problematic and misleading, as this conclusion mainly relies on correlations of single-cell outcomes related to DNA damage, p53 amplitude, and proliferation patterns within cell populations. It is important to note that characterizing a cell population as an “analog” system doesn't exclude the possibility of heterogeneous “digital” systems present within individual cells. **(B)** Moreover, it is more challenging to reveal “digital” responses using correlation analysis, especially when the variation is high (see the error bars in Figure 1G & Figure 2F-I). To claim an analog response, the authors are recommended to show the distribution of single-cell responses in their Figure 1G & Figure 2F-I (e.g., using violin plots, instead of box plots) to demonstrate graded responses in cell populations, rather than bifurcations.

We recognize that our analogy of DNA damage levels encoding into p53 and p21 signal parameters using an analog system may oversimplify the intricate mechanisms, potentially leading to misinterpretation and we apologize for that. In the new version of the manuscript, we have removed all “analog” and “digital” statements (lines 22-24 and 243-246) to ensure greater clarity in our presentation of the data.

(A) *Furthermore, we appreciate your point regarding the limitations of box plots in capturing single cell behavior and heterogeneity. In the new version of the manuscript, we incorporated supplementary violin plots in Figure 1SB and Figure 2SE and 2SI. In addition, we restructured Figure 2S to accommodate the new figures and to increase the*

comprehension and the readability of the Results Section: Gradual proliferation changes tuned with p53 and p21 amplitude.

2. (A) Conventionally, the p53 oscillator is perceived as an excitable system that maintains its amplitude regardless of the intensity of DNA damage (PMID: 21556066). However, the authors showed a dose-dependent p53 amplitude (Figure 2F & 2H). How does this phenomenon arise? **(B)** Aside from that, the authors demonstrate a measurable p53 amplitude even at zero Gy of irradiation, a scenario where one might anticipate the absence of p53 oscillations. Is the measured p53 amplitude here indicative of the post-mitotic pulse (PMID 20598361) or merely noise within the dataset? The authors should provide example traces to clarify this point.

(A) We appreciate the comments regarding the amplitude dose dependency of p53, a phenomenon that has not been described in the study cited by the reviewer. We speculate that this could be due to the distinct behavior of the chosen cell line models. Here, the amplitude effects are studied in retinal pigment epithelium cells (RPE), which might display a different range of p53 responses compared to breast cell lines such as MCF7 and MCF10. Clonal effects within cell lines might also play a role leading to different ranges of p53 responses. Moreover, a recent work studying a variety of cell lines shows a wide range of p53 responses, including amplitude variations (See PMID 28442631 Fig.3 and 7). Finally, the mechanism governing p53 amplitude dose dependency in RPE cells is undeniably intriguing, it remains enigmatic, and the experimental elucidation falls beyond the scope of our present work.

(B) We agree that RPE cells' behavior for the untreated condition differs from the expected in MCF7 cells of the mentioned study. Following the reviewer's suggestion, we have now incorporated example traces of RPE cells (Figure 2SA) into the materials provided. Representative traces show examples of sporadic and actively pulsing cells in 0Gy conditions.

3. The authors showed the inverse correlation between DNA damage and the number of cell divisions in U2OS (figure 1G), while presenting the results of p53/p21 in RPE cells (figure 2F). The observed correlation established across two distinct cell lines, might not necessarily hold true within a single cell line or consistently across various cell lines. The authors need to show the correlation in the same cell line, ideally in two different cell lines, to demonstrate the correlation holds in the same cell line and generally occurs in different cell lines.

This is an important point, and we agree that the current analysis cannot rule out that the findings might not hold true across cell lines. However, such an experiment within the same cell line will require careful engineering of new lines followed by extensive single cell measurements in hundreds of cells over multiple days. We therefore decided to strengthen the conceptual message of our study, by addressing other comments from the reviewers and including clearer statements about the limitations of our work, rather than performing additional experiments.

Following this concern, we have now added an explicit statement in the Discussion Section about the limitation of using two different cell lines and the necessity to confirm our results

within and in other lines before any generalization of our key findings is made (see lines 280-284).

4. In their Box, the authors argued that “Cells have similar intermitotic time, but the different number of division events (proliferation patterns) arises from cells entering quiescence at earlier or later time points”. To make this argument, the authors need to provide experimental evidence showing the entry of quiescence (e.g., by monitoring APC/C or CDK activity).

We appreciate and fully acknowledge the significance of the raised concern about the hypothesized scenario in which cells enter quiescence at different times. In the original publication of the RPE dataset (PMID 30057196), simultaneous single cell recordings of p21 and CDK2 activity were performed. The authors observed that p21 and CDK2 activity show an inverse proportionality, average CDK2 decreased as the levels of p21 increased (Figure 4 and S6G of PMID 30057196). This finding suggests a link between quiescence and the levels of p21 in the same RPE dataset that we studied here.

Moreover, motivated by the reviewer’s comment and the above-cited reference relating p21 and CDK2, we have now studied the median levels of p21 at the point of the last observed division. In the figure below we present the smoothed normalized median trajectory of p21 levels of untreated (0 Gy) RPE cells clustered by the total number of divisions (different colors) with the vertical lines indicating the median time of the last division of each group. This plot shows that the time of the last division is accompanied by an increase in median p21 levels. As shown in PMID 30057196, an increase of p21 correlates with a decrease in CDK2 levels, which might suggest an entry in quiescence. To verify this relationship between the last division and enter into quiescence further experiments will be needed, as suggested by the reviewer. Establishing a new RPE cell line model to monitor APC/C or CDK activity would be definitively very interesting, but it will not be feasible within a reasonable time frame, and we considered it to be beyond the scope of this manuscript. Nevertheless, following the reviewer’s suggestion we have now included on lines 149-152 and 260-261 a clarification regarding the limitation of our conclusions and how to experimentally assess both hypothesized scenarios.

Figure 1: Median of p21 levels trajectories grouped by the total number of divisions during the whole recording of untreated cells. For each group we had: 24 cells (1 division), 44 cells (2 divisions), 86 cells (3 divisions), 170 cells (4 divisions), and 50 cells (5 divisions). The vertical lines indicate the median time of the last division in each group. This figure is only for the rebuttal letter purposes.

Minor concerns:

1. Data presentation needs to be improved. For example, it is not easy to comprehend the conceptual model (Figure 1A) and the experimental results (Figure 1G & 1H), as the current visual representation does not align cohesively.

We extend our apologies for the lack of clarity in the original version of Figure 1A. In the new version, we have removed Figure 1A to avoid confusion, the text was restructured to align with the revised sequence of Figure 1 (see lines 85-86). Moreover, we adapted the supplementary Figure 1S and removed Figure 1SB, 1SC, and 1SD to increase the comprehension of Figure 1's message and to add the new violin plot Figure 1SB.

2. For most box plots, correlation coefficients are surprisingly high given how some boxes overlap quite strongly. **(A)** Which data are these coefficients calculated from (the medians only or every datapoint)? **(B)** What are the sample sizes? **(C)** The authors should clarify those points and note the necessary information in the figure legend.

We express our regret for not presenting these details in the original manuscript.

(A) *We computed the correlation coefficients using medians in order to illustrate the associations between DNA damage or p53/p21 levels and a discrete variable, specifically the total number of cell divisions.*

(B) *We now include sample sizes in the respective captions.*

(C) In the new version, we also provide comprehensive details regarding the calculation of correlation coefficients in the Methods Section (lines 304-306 and 322-323).

3. More details on analysis methods are needed. Merely referencing the Python scripts or functions used does not provide sufficient clarity for readers to assess the quality and robustness of the analysis procedures.

We apologize for any insufficiency in the information presented within the methods Section. To address this, we have made substantial additions, elaborated upon in the specified lines 309-311, 316-317, 325-326, 332-345, 348-356, and 367-368.

4. (Page 19) Figure 1A: “treshold” need to be corrected to “threshold”.

We are sorry for this mistake; in the new version, we removed Figure 1A.

5. (Page 19) Figure 1: The labeling for γ H2AX should be correct and consistent. The authors employed the label GFP-gH2AX in Figure 1D, while in Figure 1F they used γ H2AX. It is recommended to maintain uniformity in labeling throughout the study.

We appreciate the reviewer's help in pointing out this oversight and we rectified it in the new version.

Reviewer #2 (Remarks to the Author):

The manuscript by Gutu et al presented new analysis of time-lapse data of DNA damaged cells and concluded that the p53-p21 network senses the distinctive DNA damage strength and then modulates cell proliferation. Although the data are interesting, I think more experimental data and data analysis, especially those that directly link DNA damage level, p53 dynamics and cell proliferation at the individual cell level, rather than population level, are needed in order to establish the functional correlation.

It is also unclear to me what novel advance the study provides in terms of our understanding of cell fate control mediated by p53 upon DNA damage. Results from the paper appear to simply support the well-known scenario that higher DNA damage triggered higher upregulation of p53 and subsequently more cell cycle arrest (i.e., less cell division/proliferation). The authors need more single cell data to demonstrate the novel role of p53-p21 network stated in their study.

We appreciate the reviewer's thorough evaluation of our manuscript. While we acknowledge the importance of investigating the relationship between DNA damage levels, p53 activity, and cellular proliferation at the single cell level, conducting the necessary experiments would require creating a new RPE cell line model with live reporters for p53 and DNA damage, followed by tracking cell proliferation and quantifying foci in hundreds of moving live cells over several days. Unfortunately, these extensive experiments are outside the scope of our current study. In acknowledgment of the limitations of our study concerning individual cell responses, we have now duly highlighted these limitations in the discussion Section (lines 280-284).

Moreover, we understand and appreciate the concern regarding the novelty of our current study beyond the well-known fact that higher DNA damage triggers higher upregulation of p53 and subsequently more cell cycle arrest. Our study distinguishes itself from previous p53 signaling works in which it identifies the role of specific parameters of the p53 signal and describes their dose-dependent time evolution. Specifically, our unique contributions are:

1. We found that from all parameters of the p53 signal it is the amplitude of p53 that best correlates with proliferation differences among cells, indicating that cells with a lower amplitude have a greater ability to proliferate (Figure 2F-G).
2. By focusing on gradual amplitude changes we found that amplitude encodes for DNA damage strength with a gradual relationship between the damage levels and the amplitude levels (Figure 1F and 2H-I).
3. Moreover, contrary to most studies on single cell p53 signaling that use time-averaged metrics, we implemented an advanced time-series analysis technique specifically designed for noisy non-stationary data which allowed us to capture the time evolution of the p53 signal parameters (instantaneous characterization). This approach revealed a dose-dependent time-resolved behavior of the p53 period (Figure 4B). This is the first time that a parameter from the p53 signaling network is shown to evolve in time in a dose-dependent manner.
4. Moreover, our work revealed a connection between the p53 period and proliferation rate at the population level (Fig 4A). We further tracked this population effect to a subset of individual cells, where we observed that a time-localized increase in the p53 period slope coincides with cells exiting a low-proliferative into a high-proliferative state (Figure 4E-G).

Major comments:

1. **(A)** The most important data that this study needs are single cell time-lapse data that simultaneously measure the DNA damage level, p53 induction dynamics and cell fate in the same single cells. The authors should use cell lines that express both the GFP- γ H2AX reporter (shown in Fig. 1F) and a p53 fluorescence reporter (shown Fig. 2B), e.g., mRFP-p53, to acquire such data for quantitative analysis. **(B)** All the correlation analysis of DNA damage level and p53 level in the current study are done at the population level. Given the large single cell variability, such population-level statistics that average over all single cells really miss the point of single cell analysis and may not reflect the genuine correlative behaviors in individual cells.

(A) We agree on the importance of measuring DNA damage levels, p53 dynamics, and cell responses simultaneously. Previous studies used a p53 reporter together with 53bp1 fluorescent reporter for quantifying DNA damage in live cells over approximately 24 hours (PMID: 24252182). While extending this live quantification to 120 hours for RPE cell models is very interesting, it is also extremely challenging and beyond our current scope.

(B) We concur with the reviewer's concern that the correlation analyses of DNA damage levels and p53 levels being conducted at the population level might mask single cell features. We calculated correlation coefficients between two variables - the total number of divisions (represented as a vector with discrete values) and the medians of DNA damage or p53-p21 levels for each group. This approach was chosen to quantify the relationship between these

two magnitudes, one of which relies on the clustering of subpopulations by total number of divisions and therefore averaging the subpopulation within each cluster. To better represent the single cell variability within each cluster we have now included new violin plots as supplementary materials in Figure 1SB, Figure 2SE, and Figure 2SI.

2. **(A)** Moreover, the DNA damage level is obviously not at a steady-state level over time. As shown in Figure 1F (top panel), DNA damage is very high right after irradiation and then decreases over time, possibly due to DNA damage repair activated by p53. **(B)** Does this decrease of DNA damage level alter p53 dynamics (e.g., switching) and cell fate compared to the early time points? **(C)** Again this illustrates the importance of quantifying the dynamics of DNA damage level and p53 in the same cells over time, instead of simply using the average DNA damage strength as the metric to analyze the correlative behaviors.

***(A)** We apologize for the confusion arising from the misinterpretation of Figure 1F (now labeled as 1E in the updated version). Please note that the nuclear fluorescent signal shown in the live single cell imaging of Figure 1E does not report on DNA damage levels. Instead, it is a nuclear fluorescent signal used solely for the purpose of computational tracking of cells. To avoid potential misinterpretations, we have now color-coded in light blue the nuclear area of the tracked cell and included this information in the caption of Figure 1E.*

***(B)** This is a very interesting point. A previous investigation (PMID: 24252182) established a cell line that expressed both p53 and DNA damage levels. This particular study revealed an exponential decay in the number of 53bp1 foci over time with diverse decay rates (refer to Figure 4C in PMID: 24252182), while p53 signals exhibited heterogeneous dynamics. In this system, the authors observed a poor correlation between the levels of DNA damage and the dynamics of p53 pulses post-damage (see Figure 7A in PMID: 24252182). This suggests a more complex relationship between the p53 pulsatile dynamics and the DNA damage levels.*

***(C)** Certainly, the importance of having a cell line equipped with both DNA damage and p53 reporters to investigate the temporal correspondence between changes in DNA damage levels and p53 dynamics cannot be overstated, we do agree that this would be a very insightful experiment to perform but it lies beyond the scope of this work.*

(D) The authors use the intermitotic time as a surrogate for cell cycle length. But in the intermitotic time calculation, the authors excluded the very long cell cycle because no division was observed. For example, in Figure 2D, from the 3rd division to the end of the imaging experiment, as there is no 4th division that occurred, the authors called the last phase “Cell age” and excluded it from the intermitotic time statistics. I think such analysis is wrong and artificially gave a shorter intermitotic time for scenario of high cell cycle arrest, e.g., under high DNA damage. In my opinion, if the cells do not divide, it can simply mean the intermitotic time is much longer than the duration of the imaging experiment. Overall, I disagree with the authors’ use of intermitotic time as a measure of cell cycle length, and I do not think their data make a distinction between prolonged cell cycle length and some sudden decision of no division.

We agree that the experimental characterization of the IMT distributions is biased by the observation time, i.e. the only IMTs that we account for are those that happen within our

experimental recording time window. Consequently, as the reviewer clearly points out, the IMT distribution of observed division events might not accurately represent the underlying distribution of cell cycle lengths for this cell population. To avoid a potential misinterpretation of our results, we have now removed the “cell cycle length” statements when discussing IMT results and included a clarification of the intrinsic bias towards shorter IMTs in the Discussion Section. See changes in Results Section lines 105-106, 144 and 149, in the Caption of new Figure 3, and in Discussion Section lines 284-288.

4. After the cells divide, the DNA damage level, p53 and p21 level/dynamics may change in the daughter cells. The authors should quantify these features in individual cells before and after each cell division and see how these features are genuinely correlated upon division. It is possible that cell division simply acts as a dilution scheme, i.e., DNA damage is lower and p53 is lower, so it further augments the proliferative potential. Again, I think much more single cell analysis have to be performed under distinctive scenario, instead of simply using the population or time averaged values to establish the correlative features.

Certainly, we concur that investigating the impact of cell divisions on p53 and p21 dynamics is indeed very interesting. In the new version of the manuscript, we include exemplary single cell traces of p53 (Figure 2SA) with marked division events, where it can be seen that p53 levels remain relatively stable before and after a division event. Moreover, in these random examples we see that p53 and p21 levels before and after division remain relatively stable. To better quantify this, we calculated the mean levels of raw signals before and after the division events for all radiated and untreated cells and computed the correlation coefficient using the Pearson correlation (see Figure 2 below).

Figure 2: The left plot illustrates the correlation between mean p53 levels within a temporal interval spanning 2 hours before and after cellular division. Conversely, the right plot depicts the correlation between mean p21 levels during the same 2-hour timeframe surrounding cell division. Each point represents data from a single division event from a total of 3995 division. This figure is exclusively intended for inclusion in the rebuttal letter and is not meant for the manuscript.

Altogether, Figure 2 shows a good correlation between the single cell levels of p53 and p21 both before and after each cell division.

5. On page 7 (lines 197-199), distribution of the cells showing different p53 dynamics is discussed. (A) At what DNA damage level were these data obtained? (B) What are the distributions of different p53 dynamics for untreated cells and cells at other DNA damage level, e.g., 2, 4, 10 Gy? (C) It is stated in lines 209-212 that “Calculating proliferation rates differences between stables and switchers after the bifurcation time of around 60 hours indicates a dose-dependent boost effect with a 1.68, 1.84, and 4.49 fold-increase in proliferation for 2, 4, and 10 Gy, respectively”. I am confused by this statement. As DNA damage level increases, more cells should get into cell cycle arrest so the proliferation at 10 Gy should be a lot lower than that at 2 and 4 Gy. Moreover, I would think at 10 Gy, most cells should be in a long cycle arrest and do not divide. (D) What is the functional relevance and meaning of this boost in proliferation at the population level?

(A) The percentages mentioned in lines 197-199 pertain to all radiated conditions pooled together, and we sincerely thank the reviewer for highlighting the need for clarification, which we have addressed in the updated version (lines 216-219 and the corresponding caption of that figure).

(B) Thanks to the reviewer's comment we have now calculated these fractions for each condition and present the results in a new Figure 4SC in the new version of the manuscript.

(C) We apologize for our lack of clarity. Indeed, as the reviewer wrote and as shown in Figure 2C, proliferation is strongly reduced for higher radiation doses. Concerning the fold increase in proliferation within different radiation doses, it's important to note that the differences observed in the cumulative division distributions (Figure 4F) represent variations in the rate of division events between the stable and period-switchers subpopulations within each dose and do not reflect the effect between doses nor indicate the proportion of cells entering cell cycle arrest. This plot primarily reflects the relative proliferation rate differences within each dose. Furthermore, we have observed that the fraction of "stables" increases with higher radiation doses, as illustrated in Figure 4SC, especially in the case of 10Gy, where it is substantially higher compared to lower doses. To improve the clarity of our manuscript we have now added a sentence to better explain our results (see lines 232-235).

(D) This is a very interesting question and further research will be needed to understand the functional relevance. At the population level, understanding the dynamics of subpopulations might help to understand the different stages of the response to radiation in time. For example, in Figure 2C, 2 and 4Gy radiated cells show an initial growth (first ~18hs), followed by a plateau-like growth (until ~60h) and a final increase-in-growth rate (from 60hs). The transition from plateau to increase-in-growth might be related to the results presented in Figures 4E and F, but further studies are needed to characterize this potential relationship.

Minor points:

1. In Figure 1F, the labels “7 μ M” and “5 μ M” on the images should be “7 μ m” and “5 μ m”

We apologize for these mistakes and appreciate the reviewer for pointing them out. We corrected them in the new updated version.

2. Citation of “Box 1” is confusing. Can the author just rename it as a figure?

We regret the confusion caused by the naming of the Box Section. As a result, we have modified its label to "Figure 3" in the revised edition and implemented the following changes: dividing the figure into subfigures (A, B, C, D, E, and F), re-editing the captions Section and add the corresponding explanations in the new version of the main text (lines 109-118, 121-124, and 149-152). To improve the clarity of the sketches we removed the figure of the distribution of the total number of divisions that was below the sketch of individual proliferation patterns.

Reviewer #3 (Remarks to the Author):

This very interesting work by Gutu et al. explores the relation between DNA damage, the dynamic response of p21 and p53 activities, and proliferation patterns at the single cell level.

Through systematic analysis, the authors uncover a gradual relation between DNA damage, cell proliferation, and key features of p53 and p21 activity. They show changes in the amplitude of p53 and p21, as well as alterations in the inter pulse interval of p53, encoding damage strength and shifts in proliferation pattern. Finally, the analysis reveals a temporal switch in p53 oscillatory period, that correlates with a subpopulation of cells transitioning from low to high proliferation.

In my opinion this is an excellent work, a very rigorous study addressing relevant questions about how proliferation heterogeneity is produced in a population of cells. The data analysis is of very high quality and leads to novel conclusions and ideas that may be of broad interest. I have only a few comments for the authors to consider.

We deeply appreciate the encouraging feedback and have implemented the recommended enhancements to improve our manuscript.

Comments.

1. Fig. 1H: You conclude that there is a poor correlation between DNA damage levels and inter-mitotic times. Not sure I got this right, in this scatter plot you don't see cells with large DNA damage levels and long inter-mitotic times. Does the inverse of DNA damage levels correlate with inter-mitotic times?

We are sorry for not being clear enough regarding Figure 1H (updated label Figure 1G). Here, we calculated the intermitotic times only for actively dividing cells. Consequently, the cells that entered a state of cell cycle arrest do not feature in this plot. In other words, for the cells remaining in an arrested state throughout the experiment, we were unable to determine intermitotic times (IMT). These cells were characterized by notably elevated DNA damage levels. However, the cells that were actively dividing had a relatively constant intermitotic time (see Figures 1G and 3). In particular, these cells did not exhibit a correlation with the DNA damage levels.

2. Box: this result is important. Should it be in a Figure then? I understand that it contains data relevant to both Figs. 1 and 2, so maybe you could split this.

We appreciate the reviewer's acknowledgment of the importance of the results presented in the Box Section. In response, we have updated its designation to "Figure 3" in the revised

version and added the corresponding explanation and captions in the main text as lines 109-118, 121-124, and 149-152.

- 3.** Fig. 2 and related text. **A)** The definition of amplitude was not clear to me. Is it the total amplitude of the signal, that is its absolute height? Or is it the amplitude from trough to peak values?
-B) Also, it is not clear what is correlated in Fig. 2E: In Fig. 2D you show a continuous amplitude envelope, as well as a continuous trend. **C)** These are curves, but then they are correlated to some scalar values like IMT (I assume this is the average of all IMTs of a given cell?). **D)** Is the average amplitude considered here? **(E)** Reference to modules or functions from pyBOAT are given in the methods, but still I think that these things need some clarification in the text.

***(A)** We regret the lack of clarity in our quantification of the amplitude shown in Figure 2. For the amplitude quantification, we first identify the higher power main oscillatory spectral component using continuous wavelet transform as implemented in the pyBOAT software package for Python (v0.9.2). From this main oscillatory component, detected by spectral ridge-detection, we use the wavelet power spectrum to estimate the amplitude using the Morlet Wavelet scaling factor (see: "Optimal time frequency analysis for biological data - pyBOAT"). The amplitude determined through wavelet analysis exhibits a good correlation with that obtained through a peak-picking approach. Following the reviewer's comments we have now included a more comprehensive description in the Results and Methods Section (lines 159-162 and 332-345) and Figure 2SC to consider the similarity in amplitude derived from pyBOAT and the peak peaking method in Scipy.*

***(B)** As for Figure 2E, we have correlated the values of the area under the curve (AUC), the amplitude, the standard deviation, the period, the intermitotic times, the cell age, and the total number of divisions. For that, we computed time-averaged median values from continuous curves such as the amplitude. When it comes to the area under the curve (AUC), we determined it by averaging the results obtained from two methods, namely the trapezoidal and Simpson methods. We have now added a clearer description about these calculations in lines 348-356.*

***(C)** For the intermitotic time (IMT) we considered the mean IMT per cell. We have now included a more explicit clarification on how this is calculated in lines 348-349.*

- 4.** From the abstract and introduction, I got the impression that it was an experimental study. However, as far as I can understand, it relies on data analysis from previous work: - Fig. 1 data: To evaluate how individual cells' proliferation activity relates to DNA damage levels we build up and expanded the analysis of a dataset from our group, originally published in the supplements of [12]. - Fig: 2 data: For this, we analyzed one of the longest published 125 datasets of individual human cells treated with a gradient of radiation-induced DNA damage, see (Figure 2A) and [14]. Still, in the Methods there is a section about U2OS cells. Are the experiments in the Box new? I think this should be stated more clearly to avoid wrong expectations.

We regret the miss-impression that can derive about the nature of our work. We have now included explicit references indicating that our work relies on data analysis from previous works. This is now corrected in the Abstract (line 19-20), the Introduction (lines 69-70), and in the Methods Section

Minor points.

5. Fig. 1 A, maybe it would be more clear to plot the single cell behavior on the left for both types of population responses. At the moment it looks as if the left panel applies to both cases, but for the heterogeneous case the single cell behavior is plotted in the "Population" panels on the right. - Fig. 1A, top right panel, I think it should be a solid line (like in the bottom panel) instead of dots, to make it clear it is the same thing, that is the population response. - Fig. 1A, bottom right panel, I can barely see the colored response curves, can you make them thicker?

We apologize for any confusion arising from this figure. Based on yours and Reviewer 1's comments we decided to remove this subfigure and its references, which we think will improve the readability of our manuscript.

6. Fig. 1F is not mentioned, it should come on page 4 together with 1G?

We have now corrected this missing reference to Figure 1F in lines 97-104 (which is now Figure 1E in the new updated version).

7. Fig. 2A: I think RPE cells acronym is not defined.

Thanks for the comment. We now added the definition of the acronym RPE in lines 12-14 in the Abbreviations Section. We also added the definition in the main text (line 135).

8. Fig. 2E: the text labels of correlations are tiny and cannot be resolved in normal size.

We apologize for compromising the readability of the figures and we have now edited all figures and increased the size of the labels in the new version.

9. page 6, line 178 - 180: "This classification indicated that highly proliferative cells pulse on average with a significantly lower period than their lower proliferative counterpart..."
This should be ... slower period...?

Thanks for the remark. We have now changed the term for the adjective longer in lines 197-199.

10. Augmented Dickey-Fuller test: the spelling is wrong (and mutating) in the text (line 187), in the Methods (line 344), and in Fig. S3B legend.

We regret the spelling mistake. Now we use the correct name "Augmented Dickey-Fuller" in the updated version of the manuscript (lines 393) and captions of Figure 4SB.

REVIEWERS' COMMENTS:

Reviewer #1 (Remarks to the Author):

The authors have addressed most of the questions I raised.

Reviewer #3 (Remarks to the Author):

The authors have addressed my comments and questions thoroughly, and revised the manuscript accordingly.